# Headache Prevalence and Its Associated Factors in Makkah, Saudi Arabia

**DOI:** 10.3390/biomedicines11102853

**Published:** 2023-10-20

**Authors:** Maram H. Alshareef, Bayan Hashim Alsharif

**Affiliations:** 1Department of Community Medicine and Pilgrims Health, Faculty of Medicine, Umm Al-Qura University, Makkah 21955, Saudi Arabia; 2Hajj and Umrah Research and Epidemiology Administration, King Abdullah Medical City, Makkah 24331, Saudi Arabia; alsharif.b@kamc.med.sa

**Keywords:** epidemiology, headache, migraine, pain, prevalence

## Abstract

Primary headaches are more prevalent and associated with several risk factors, such as chronic diseases, unhealthy lifestyles, smoking, caffeine intake, work, and stress. However, these factors are not associated with specific headache disorders. We investigated the prevalence of primary headache disorders and the associated risk factors in Makkah. This cross-sectional study, conducted over a 6-month period, used an anonymous survey disseminated through online platforms. The questionnaire was a modified version of a validated questionnaire used to assess headaches in relation to modifiable and non-modifiable risk factors. In total, 1177 participants aged 18–65 (mean, 31.5 ± 12.6) years were included. Headaches were diagnosed among 44.2% of participants aged 20–59 years, with a high prevalence among young adults; additionally, 71.5% of participants with headaches reported experiencing headaches for <15 days per month. Chronic headaches were found in 28.5% of participants. Age, chronic diseases, work, caffeine consumption, and smoking were associated with having a significant effect on headache prevalence. The prevalence of headaches in Makkah has increased in comparison to that reported in previous studies. Certain modifiable and non-modifiable risk factors have been associated with headaches. Headaches impact all life aspects of individuals and communities. An educational program for professionals and patients can improve patient outcomes.

## 1. Introduction

Headache disorders are a common problem worldwide [1]. These disorders affect individuals of all age groups and both sexes. Headaches can be a major burden on countries as they affect individual health and governmental economies [2], with headache disorders being the second cause of disabilities worldwide [3]. Headaches can cause physical, social, psychological, and economic impacts on individuals and communities [4,5]. The International Classification of Headache Disorders (ICHD) divides the cause of headaches into primary and secondary disorders. Primary disorders, as diseases themselves, include migraine headaches, tension-type headaches, and autonomic cephalgia headaches, such as cluster headaches and hemicrania continua. In contrast, secondary headaches are symptoms of other diseases that could require emergency care, such as infectious diseases (e.g., meningitis and encephalitis) or cerebrovascular diseases (e.g., hemorrhagic strokes and aneurysms). Other types of headaches might require urgent care management similar to what needs to be undertaken in cases of sinusitis, otitis media, or temporomandibular joint disorder [6]. Many of these types of headaches have an unknown prevalence in the community and some previous studies have indicated a lack of knowledge concerning different types of headaches and the misuse of diagnostic guides among general practitioners [7,8,9].

However, several cross-sectional studies have found that headache prevalence ranges from 28.8% to 91.3%, with an overall global prevalence of 52% [2,10,11]. In Saudi Arabia, a cross-sectional survey in 2020 showed that the prevalence of headaches was approximately 65.8%, of which a higher prevalence of primary headache disorders was observed [12]. Recent global and national studies revealed that migraines had a prevalence of 14–37.9% and 4.9% of those diagnosed with migraines suffered from the consequences of the disability [13,14]. In addition, cluster headaches have a prevalence of 0.1–1.3% [15].

In 2020, the International Association for the Study of Pain stated that pain is an individual experience based on physical, psychological, and social interactions [16]; therefore, several risk factors can be associated with primary headache disorders and precipitate headache onset. These factors include sex, lifestyle, marital status, working hours, sleep deprivation, chronic diseases, psychological disorders, and the excessive use of over-the-counter analgesic medication [11,17,18,19].

Headaches can affect all aspects of a person’s life, including physical and psychological well-being, which can lead to chronic headaches, chronic diseases, and psychiatric disorders, such as depression, anxiety, stress, and social isolation. These adverse health effects are expected if the headache disorder is not diagnosed and treated appropriately [20]. Frequent headache attacks can precipitate the misuse of over-the-counter medications that lead to rebound headaches. In addition, they can cause physical, social, and psychological stress that increase the prevalence of chronic headaches. Medication misuse and chronic headaches can be prevented if acute headache attacks are diagnosed and treated early [21]. Despite the importance of the psychological management of headaches, there is a lack of psychological support centers for people with chronic headaches [22].

Most primary headache disorders can be managed in the community setting through medications that are easily accessible. However, general practitioners need knowledge and experience on when and how to prescribe abortive and preventive medication [7,23]. Furthermore, a general practitioner can diagnose and manage most of the headache disorders by following guidelines from the ICHD and other headache institutes [24,25]. However, a previous study showed that general practitioners often lack experience and confidence in headache diagnosis, as well as knowledge regarding headache management [26]. Consequently, headache management has been negatively influenced by factors such as physicians’ experience, patient compliance, system policies, and lack of medication, which can be easily overcome to improve the patients’ quality of life [27].

The city of Makkah has a population of approximately 2,079,000 people [28]. A recent study conducted in Makkah revealed that the headache prevalence among Makkah population was 8.9%, which was nearly comparable to that of other studies conducted among the general population in Riyadh, the capital city of Saudi Arabia [12,29]. Most individuals seek medical advice in public primary care centers, which have limited facilities. Moreover, there is a lack of headache centers in Makkah to cover the needs of the residents. Therefore, the exact prevalence of headaches in the general population is unknown because not all individuals who experience headaches seek treatment. This study aimed to estimate the prevalence of headache disorders in association with several personal, social, and lifestyle factors that affect it. Understanding the magnitude of the problem may guide the establishment of a program that can improve the outcomes for this population.

## 2. Materials and Methods

This study focused on individuals living in Makkah. Makkah is a city with a significant history spanning back centuries. It is the holiest city for all Muslims. The people in Makkah are Saudis from several cultural backgrounds and this has varied over decades as pilgrims visited Makkah during the Hajj season. Health facilities in Makkah are provided by the public and private sectors. Approximately 10 hospitals are available to serve the population in Makkah. One of them is a tertiary hospital that has a pain center, which can provide chronic headache management. Otherwise, cases of headaches are treated in the neurology clinic if the patient is referred by the PHCP; however, the referral can be delayed when the PHCP fails to comprehend the magnitude of the problem. We conducted this study to calculate the prevalence of primary headache cases missed or not referred to other healthcare facilities according to the diagnosis of the primary healthcare provider (PHCP). This was a cross-sectional study performed through an anonymous survey that was communicated over a 6-month period. In this work, a voluntary response sampling technique was utilized. The questionnaire was distributed electronically through emails and other social media platforms.

The minimum sample size required for this study was calculated using OpenEpi (version 3.0) with the following considerations: the population size is approximately 2,115,000 and the confidence interval (CI) level of 95% is maintained. Therefore, the sample size needed was calculated to be 385 participants.

A modified structured questionnaire from the previous study was used and distributed anonymously through data collectors on online platforms [30]. The questionnaire included four sections. The first section included questions concerning the demographic characteristics, namely, age, sex, marital status, education level, income, and chronic disease. The second section included questions regarding work-related factors. The third section included questions concerning lifestyle habits. The fourth section included questions regarding headache history, diagnosis received, onset, type of headache, duration, number of onsets, and use of over-the-counter analgesics.

Bioethical approval was obtained for this study from the institutional research ethics board of Umm Al Qura University.

Data were analyzed using SPSS software (version 22; IBM Corp., Armonk, NY, USA). Numerical data were presented as means ± standard deviations or as medians, depending on the type of distribution of each variable. Percentages were used for categorical variables. Chi-squared tests were performed to compare the categorical values.

## 3. Results

In total, 520 (44.2%) participants reported frequent headaches while 657 (55.8%) did not report headaches (Figure 1).

A total of 1177 participants were enrolled in this study, of whom more than half were females (890, 75.6%) between 18 and 65 years old (Table 1). The mean age of participants was 31.5 ± 12.6 years. Some sociodemographic factors were shown to be significantly associated with headaches; these included age, marital status, and chronic disease (Table 1). In addition, headaches showed a significant association with the 40–59 year age group (*p* = 0.001) and chronic disease (*p* = 0.001); moreover, headaches were significantly associated with married or divorced and widowed individuals (*p* = 0.001) (Table 1).

Although there was no significant difference between sex and nationality and the prevalence of headaches, 48.1% of males reported headaches and 49.3% of non-Saudis reported headaches (Table 1).

Headache prevalence was significantly associated with working status; half of the working participants reported headaches (*n* = 288, 51.2%; *p* = 0.001) and more than half of the non-healthcare workers reported headaches (*n* = 303, 61.1%; *p* = 0.001) (Table 2). In contrast, participants who reported more work hours exhibited a higher prevalence of headaches. This was observed in participants who worked multiple jobs (*n* = 65, 58%) and >12 h/day (*n* = 61, 51.3%); however, these results were not significant (Table 2).

Lifestyle was shown to greatly influence headache prevalence. Smokers (*n* = 119, 54.3%) and ex-smokers (*n* = 53, 61.6%) experienced more headaches than non-smokers (*n* = 345, 39.9%; *p* = 0.001) (Table 3). Despite the significant association observed between caffeine consumption and headache prevalence (*p* = 0.006), more than half of participants who drank caffeine (*n* = 479, 53.6%) did not report headaches; however, the amount of coffee consumed seems to be associated with headaches (with 37.1% of non-caffeine drinkers, 43.8% of those drinking 1–2 cups per day, 50.0% of those drinking 3–4 cups per day, and 57% of those drinking >4 cups per day reporting headaches) (Table 3).

Logistic regression analysis was conducted to examine the relationship between sociodemographic, work-related, and lifestyle variables and the likelihood of experiencing a headache.

The aforementioned factors were analyzed using multivariate logistic regression analysis (Table 4), which showed that having a history of chronic disease and coffee drinking remained an independent risk factor that led to the diagnosis of a headache. Participants with chronic illnesses were more likely than normal individuals to visit a healthcare provider and be diagnosed with a headache (OR, 2.580; 95% CI, 1.718–3.874; *p* = 0.000). Individuals who regularly consumed coffee were more likely to experience headaches compared to those who did not drink coffee regularly (OR, 1.800; 95% CI, 1.218–2.661; *p* = 0.003).

Our findings showed that non-smoker status, the use of non-steroidal anti-inflammatory drugs (NSAIDs), the use of paracetamol, and a relatively short duration of using non-prescribed analgesics (<3 months) were associated with increased protection against headaches.

Non-smokers were more likely to be protected against experiencing headaches than ex-smokers (OR, 0.526; 95% CI, 0.282–0.982; *p* = 0.044). Individuals who used NSAIDs were three times more likely to be protected against experiencing headaches than those who used other types of analgesics (OR, 0.032; 95% CI, 0.003–0.353; *p* = 0.005). Participants receiving paracetamol were more likely to be protected against experiencing headaches than those who did not receive paracetamol (OR, 0.466; 95% CI, 0.260–0.836; *p* = 0.010).

Those with a shorter duration of using non-prescribed analgesics were more likely to be protected against experiencing headaches compared to those with a longer duration of use (OR, 0.680; 95% CI, 0.491–0.942; *p* = 0.020) (Table 4).

## 4. Discussion

In this study, we aimed to examine the prevalence of primary headache disorders and the associated risk factors in Makkah, Saudi Arabia. The prevalence of headache disorders is increasing worldwide and almost half of the world’s population is currently affected [2]. The impact of headaches needs to be continuously assessed and managed, owing to their effect on individuals and economic health expenditures [31,32]. The prevalence of headaches ranges from 28% to 91.3% in various countries [2,10,11], with an average of 52% worldwide [2]. The prevalence of headaches in the current study was 44.2%. This result is different from previous studies performed in several cities in SA, one of them among university employees and students in Riyadh, which found the prevalence was about 82.12% [33]. In contrast, Almalki et al. showed that the prevalence of headaches among Al-Kharj city residents was 3%, which could be due to the fact that the participants in that study were selected from an educated population in specific sectors [34]. Moreover, a study including all of Saudi Arabia observed that the 1-year prevalence of all headaches was 65.8%. Furthermore, the prevalence of migraines and tension-type headaches (TTHs) were 25.0% and 34.1%, which were near the prevalence reported in the current study, 31.7% and 25.2%, respectively [12]. On the other hand, migraines and TTHs constituted 31.7% and 25.2% of the headaches reported in the current study, respectively. This minimal difference may be attributed to misdiagnosis by healthcare providers or misperception by patients because patients were asked about the diagnosis they received from the HCP in this study [35]. This possibly highlights the inadequate professional health information concerning headaches and their different diagnoses, the lack of use of the International Classification of Headache Disorders-3, which was noticed in a few previous studies, plus, the lack of knowledge regarding preventive measures that can reduce headache attacks [7,23,26]. There is a need for greater professional education and health awareness to overcome the misdiagnosis of headaches and provide satisfying healthcare to patients who suffer from headaches [6,27]. Additionally, more health education regarding headaches and their types can enhance patients’ understandings of their condition. Despite the high prevalence of headaches, no health awareness campaign is conducted yearly to increase awareness about this global condition in Saudi Arabia in alignment with the ICHD organization [8,9]. Although a headache is an idiopathic disorder, certain risk factors can aggravate or precipitate the onset of headaches. Younger participants were more prone to headaches (*p* < 0.001); this was similar to the findings of a previous study in which younger age was a risk factor for headaches [11]. This can be explained by the stress that the younger population is currently facing and the pressure arising with the pace of life in addition to multiple distractors that affect their daily life [36,37]. Marital status was also shown to be significantly associated with headache prevalence in the current study; married participants had a significantly lower prevalence of headaches than single participants (*p* < 0.001). This result was similar to a previous study, which found that participants who were single had a higher risk of headache symptoms than married participants in the city of Al-Kharj [34]. In contrast, a study conducted in 2017 on migraines found that the number of migraine attacks can affect the rate at which couples break up. The number impacted 8% of break ups, with more frequent headache attacks per year. In contrast, the same study showed migraines became more stable with a happy marriage where more than half of the participants were happy with their partners, which indicated that marriage can be a supportive factor for patients with headaches if the spouse is supportive [38]. Moreover, the presence of chronic diseases was found to be significantly associated with headache prevalence (*p* < 0.001). However, some studies have found an association between headaches and chronic diseases, such as diabetes and hypertension, while other studies have not [34,38,39]. A recent study surveyed the American population to investigate the presence of chronic diseases with migraine headaches; it demonstrated that people who suffer from migraines have a higher chance of having other comorbidities, such as stroke, high blood pressure, angina, gastrointestinal disorders, psychiatric disorders, and insomnia. Additionally, chronic headaches can also weaken the immune system as frequent headache attacks can lead to more autoimmune diseases and allergic reactions, such as asthma and urticaria [40]. Specifically, chronic diseases can cause psychological stress and vice versa. Additionally, a strong association between them has been reported in several studies [41,42]. Work was another factor assessed in the current study for its association with headaches. The absence of headaches in participants was 48.8% in working participants and 62.2% in those who were not working; this showed a significant association between work status and headache prevalence. This is supported by a previous study conducted in China, which showed that work could affect the prevalence of headaches [17]. Additionally, in the current study, being a worker showed a significantly higher association with headaches than not being a worker, which was documented in a previous study showing that working status can aggravate headache onset [34]. However, previous studies have shown that female sex, higher position, sleep deprivation, and work type have a strong association with metabolic syndrome and headaches among workers. This resulted in a higher psychological impact owing to headaches, where depression and anxiety were found to have higher incidents among workers with severe headaches [43]. Moreover, findings from a previous systematic review showed that students in Bangladesh were at a high risk for developing headaches, which does not agree with our finding regarding the prevalence of headaches among students [37]. The following factors may increase the onset of headaches among students: social and academic stress, grade performance, sleeping disorders, family income, unhealthy lifestyle habits (i.e., lack of exercise), eating unhealthy food, and long hours of electronic use in addition to psychological issues [36,37]. The prevalence of headaches at a younger age is considered a risk factor for developing headaches during adult life [44]. In this study, other well-known factors that showed a significant association with headaches were smoking (*p* < 0.001), caffeine intake (*p* < 0.006), and the number of caffeinated cups consumed (*p* < 0.03), in agreement with the findings of previous studies [34,39]. Despite the debate on this result, a previous literature review showed an insufficient association between caffeinated beverage intake and headaches. Although some studies have shown that caffeinated beverages can trigger migraine attacks [18], other studies have reported that it is safe and effective to treat acute migraines with caffeine. Moreover, few studies have shown that it is safe to use caffeine for the treatment of acute headaches in combination with other types of analgesics for an effective means to abort headaches [18]. Interestingly, the results of this study showed another significant association between caffeinated beverage consumption and headache prevalence; however, the number of cups of coffee consumed was the reason for obtaining variable results [45]. Concerning smoking, a previous study conducted in Korea showed a significant relationship between smoking and cluster headache severity. It showed that patients who had never been smokers had less severe attacks and patients who smoked had severe attacks with longer durations and higher frequencies [46]. The intensity of headaches reportedly increases with the daily consumption of cigarettes and the number of smoking years [47]. These results highlight the warning risk of headaches among individuals who smoke and drink caffeine and are currently students; all of these features are associated with young age. This means that there is a higher possibility for more future cases of headaches and the development of their complications, such as having more chronic headaches owing to medication overuse. Similarly, prolonged use of over-the-counter medications was found to be associated with a higher duration of headaches [48]. In contrast, using analgesics of any type for acute headache attacks for a short duration was found to abort the attack and work as a protective factor against developing a chronic headache. It is known that appropriate treatment of acute pain can prevent the progression of pain from acute to chronic [21]. Consequently, this will reduce the physical, social, and psychological impacts that affect individuals and communities and the impact on the economy [4,5]. This finding must be taken into consideration and serious action must be taken to reduce the risk factors associated with headache development, especially among the younger population, to further reduce the impact of this global problem.

Despite the significant results of this study, it has some limitations. Although this study presented the patients’ understanding concerning the presence of headaches, it showed the limited education provided by healthcare providers and the limited understanding of the participants concerning their headache diagnosis and management. Moreover, data collection was conducted through personalized data collectors from the general population to achieve a better understanding regarding population perception and reduce selection bias. However, financial constraints and the lack of accessibility for electronic collection led the investigator to use this technique. Future studies comparing results collected from a community healthcare setting can reduce bias and potentially produce better results.

## 5. Conclusions

Headaches are a major health problem worldwide. Understanding the prevalence of headaches in communities can direct health leaders to make healthcare more accessible for these individuals. In Makkah, headache prevalence was 44.2%, which is considered high and is higher than other common health problems, such as diabetes and hypertension. Many risk factors, such as lifestyle, social characteristics, or previous health conditions, are known to be associated with headaches. In this study, we found that age, marital status, presence of chronic disease, work status, being a student, smoking, and drinking caffeinated beverages are significantly associated with headache disorders in this community. This warrants more attention from healthcare providers to obtain detailed histories from patients to appropriately diagnose and treat headaches to prevent further consequences, such as developing chronic headaches, stress, and medication overuse for headaches. There is a lack of knowledge concerning diagnosing and managing this global problem and the appropriate time required for secondary referral. There is an urgent need for improved professional education and health awareness programs should be organized to improve the diagnosis, treatment, and outcomes of patients with headaches.

## Figures and Tables

**Figure 1 biomedicines-11-02853-f001:**
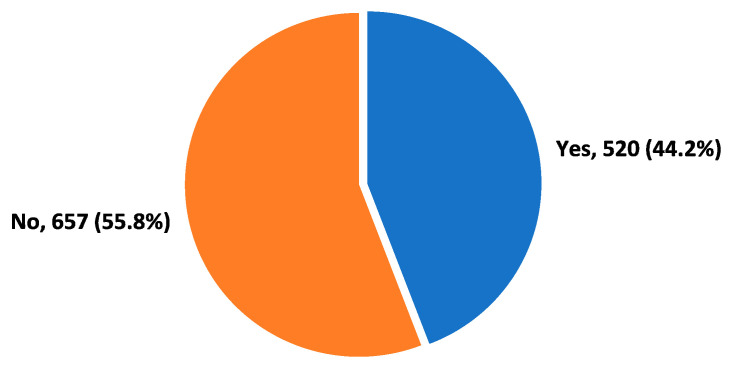
The prevalence of headaches among the study participants.

**Table 1 biomedicines-11-02853-t001:** Association between headaches and sociodemographic characteristics (*n* = 1177).

Sociodemographic Data		Total		Reported Headache				*p*-Value
				Yes		No		
		No.	%	No.	%	No.	%	
Age (years)	<20	167	14.2	71	42.5	96	57.5	0.001 *
	20–39	666	56.6	244	36.6	422	63.4	
	40–59	295	25.1	180	61.0	115	39.0	
	≥60	49	4.2	25	51.0	24	49.0	
Sex	Male	287	24.4	138	48.1	149	51.9	0.126
	Female	890	75.6	382	42.9	508	57.1	
Nationality	Saudi	1039	88.3	452	43.5	587	56.5	0.200
	Non-Saudi	138	11.7	68	49.3	70	50.7	
Marital status	Single	598	50.8	200	33.4	398	66.6	0.001 *
	Married	477	40.5	259	54.3	218	45.7	
	Divorced/widowed	102	8.7	61	59.8	41	40.2	
Education level	Below secondary	28	2.4	15	53.6	13	46.4	0.597
	Secondary	315	26.8	139	44.1	176	55.9	
	University/above	834	70.9	366	43.9	468	56.1	
Income	Sufficient	714	60.7	300	42.0	414	58.0	0.063
	Insufficient	463	39.3	220	47.5	243	52.5	
Chronic diseases	Yes	288	24.5	196	68.1	92	31.9	0.001 *
	No	889	75.5	324	36.4	565	63.6	

* Chi-squared test.

**Table 2 biomedicines-11-02853-t002:** Association between headaches and work-related factors.

Bio-Demographic Data		Total		Reported Headache				*p*-Value
				Yes		No		
		No.	%	No.	%	No.	%	
Currently working	Yes	563	47.8	288	51.2	275	48.8	0.001 *
	No	614	52.2	232	37.8	382	62.2	
Recent work dismissal	Yes	45	8.0	28	62.2	17	37.8	0.122
	No	518	92.0	260	50.2	258	49.8	
Job title	Student	429	42.1	120	28.0	309	72.0	0.001 *
	Non-healthcare worker	496	48.7	303	61.1	193	38.9	
	Healthcare worker	93	9.1	46	49.5	47	50.5	
More than one job	Yes	112	19.9	65	58.0	47	42.0	0.104
	No	451	80.1	223	49.4	228	50.6	
Daily work/study hours	<8 h	543	46.1	237	43.6	306	56.4	0.256
	8–12 h	515	43.8	222	43.1	293	56.9	
	>12 h	119	10.1	61	51.3	58	48.7	

* Chi-squared test.

**Table 3 biomedicines-11-02853-t003:** Association between headaches and lifestyle.

Bio-Demographic Data		Total		Reported Headache				*p*-Value
				Yes		No		
		No.	%	No.	%	No.	%	
Smoking	Smoker	219	18.6	119	54.3	100	45.7	0.001 *
	Non-smoker	872	74.1	348	39.9	524	60.1	
	Ex-smoker	86	7.3	53	61.6	33	38.4	
Sleep hours	<7 h	792	67.3	364	46.0	428	54.0	0.078
	>8 h	385	32.7	156	40.5	229	59.5	
Caffeine consumption	Yes	894	76.0	415	46.4	479	53.6	0.006 *
	No	283	24.0	105	37.1	178	62.9	
Cups of caffeinated drinks per day	1–2 cups	605	68.0	265	43.8	340	56.2	0.033
	3–4 cups	192	21.6	96	50.0	96	50.0	
	>4 cups	93	10.4	53	57.0	40	43.0	

* Chi-squared test.

**Table 4 biomedicines-11-02853-t004:** Multivariate logistic regression results for headaches and sociodemographic characteristics.

	B	Sig.	Exp(B)	95% CI for Exp(B)	
				Lower	Upper
Age (≥60 years)		0.064			
Age (<20 years)	0.902	0.079	20.464	0.900	60.750
Age (20–39 years)	0.396	0.400	10.486	0.590	30.743
Age (40–59 years)	0.699	0.135	20.012	0.805	50.028
Sex (Male)	−0.260	0.193	0.771	0.522	10.140
Nationality (Saudi)	0.120	0.638	10.127	0.684	10.858
Marital status (Divorced/Widow)		0.183			
Single	−0.407	0.251	0.665	0.332	10.334
Married	−0.016	0.962	0.984	0.509	10.904
Education		0.817			
Below secondary	0.459	0.539	1.582	0.366	6.832
Secondary	−0.021	0.919	0.979	0.656	1.462
Income (Sufficient)	−0.049	0.764	0.952	0.693	1.308
Diseases (Chronic diseases)	0.948	0.000 *	2.580	1.718	3.874
Participants working status (Working)	−0.210	0.261	0.811	0.563	1.168
Job title (Healthcare worker)		0.001 *			
Student	−0.433	0.169	0.649	0.350	1.203
Non-healthcare worker	0.447	0.107	1.563	0.908	2.693
Smoking (Ex-smoker)		0.075			
Smoker	−0.331	0.335	0.718	0.366	1.409
Non-smoker	−0.642	0.044 *	0.526	0.282	0.982
Sleeping hours (<7 h)	0.287	0.106	1.332	0.941	1.885
Drinking coffee	0.588	0.003 *	1.800	1.218	2.661
Using unprescribed analgesics for headaches	0.233	0.164	1.262	0.909	1.752
Analgesic type		0.005 *			
NSAIDs	−3.434	0.005 *	0.032	0.003	0.353
Paracetamol group	−0.763	0.010 *	0.466	0.260	0.836
Muscle relaxant	−0.357	0.539	0.699	0.224	2.186
Others	0.049	0.930	1.050	0.356	3.098
Duration of using unprescribed analgesics (<3 months)	−0.386	0.020 *	0.680	0.491	0.942
Source of analgesics (My own self)		0.023			
Physician	0.969	0.015 *	2.635	1.209	5.743
Pharmacist	−0.077	0.686	0.926	0.640	1.342
Family/friends	−0.361	0.087	0.697	0.461	1.054
Brochure	0.302	0.513	1.352	0.547	3.339

* *p* < 0.05.

## Data Availability

The data presented in this study are available on request from the corresponding author.

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
