# Peer review of "Headache Prevalence and Its Associated Factors in Makkah, Saudi Arabia"

_biomedicines, 2023, doi:10.3390/biomedicines11102853_

Round 1

Reviewer 1 Report (Previous Reviewer 2)

In this version the paper is suitable for publication

Author Response

Dead reviewer, 

thank you very much for the great support and valuable comments that were helpful to improve my manuscript and product it and best publishable form 

Regards 

Reviewer 2 Report (Previous Reviewer 3)

Line 147 is still wrong "Although there was no significant difference between sex and nationality and the prevalence of headaches, of the participants that reported headaches, 48.1% were male, and 49.3% were non-Saudis (Table 1)". There were 520 subjects with headache (Line 133 and by addition in table 1). From Table 1, 138 males reported headache and 68 non-Saudi reported headache. The correct statement would be either:

"Although there was no significant difference between sex and nationality and the prevalence of headaches, 48.1% of males reported headaches, and 49.3% of non-Saudis reported headaches (Table 1)."

or (because 138/520 =26.5% and 68/520 = 13.1%)

"Although there was no significant difference between sex and nationality and the prevalence of headaches, of the participants that reported headaches, 26.5% were male, and 13.1% were non-Saudis (Table 1)".

SPSS will give an accurate answer but you must then express the answer in correct English or the meaning is lost.

See above

Author Response

Dear Reviewer, 

I appreciate all your effort and the insightful comments given to improve the quality of the manuscript that Help to product it in the best publishable form.

Regards 

This manuscript is a resubmission of an earlier submission. The following is a list of the peer review reports and author responses from that submission.

Round 1

Reviewer 1 Report

Dear authors, I read with interest your study, which provides valuable information about the headaches and its associates factors in Makkah, Saudi Arabia. A series of formal aspects should be addresses, and my main comment is  following a reporting guideline to ensure that you covered all relevant aspects. This should be sufficient to reproduce you study, or to combine anmd meta-analyze the data in future studies.

Background: the introduction could be abbreviated. For this, take into consideration the journal s author s instructions, where general statements about headache are discouraged, since most readers will be quite aware of headache classification, and better focus on the possible research gaps.

Please, end the background section stating the study objectives. 

Methods: line 92 the target population was people living in Makkah City- provide more characteristics of the population interest: distribution of age, gender, educational levels and socioeconiomics status, proportions living in rural and urban areas, etc.

Such knowledge allows evaluations of representativeness of the participating sample, and statistical adjustments with regard to these features when necessary.

Line 100: the method of enquiry should be adequately described, Reasons of non- participation should be recognized. 

Line 106: Headache caseness should be defined precisely. It is not clear how headache type was diagnosed. How did you handle the possibility of having both TTH and migraine? Is this perhaps chronic migraine?

Line 114-128 seems to be out of topic.

Reference to the STROBE statement is recommended. 

Results: You may consider creating a table to presents results that report the headache type prevalence of migraine, TTH and other headaches types have diagnosed through the fourth section of the questionnaire.  

Discussion: In the discussion you should ideally include an explanation about the findings of your study that differ from the existing literature,

You should acknowledge and explain the possible limitation of your study. These should include improvements in the methodology that could affect the internal validy and how generizable your findings are. 

Author Response

replay attached in the word document 

Reviewer 2 Report

It is an interesting work that tries to deny the knowledge of the prevalence of headaches in a specific area. There are some recommendations that could make the article easier to read and understand.

Lines 133 and following in table 1-2-3= What does the last p-value column indicate in tables? Specify better.

Table 3, last item =  This assessment is very rough and not very specific. It does not allow for scientifically significant conclusions to be drawn. The works cited in the  references underline the ambiguity of the results in this topic and the possibility that caffeine is a possible therapeutic attempt to improve the analgesic action of numerous pian killers. Therefore this aspect needs to be reconsidered in the discussion section and rewritten.

Lines 173-174= This statement accumulates the prevalence of migraine and tension or secondary headaches, which have different pathogeneses and different therapies and therefore cannot be combined in the same evaluation. This sentence should be deleted or discussed appropriately

Finally, the discussion is too long, confusing and does not follow a logical scheme, but proposes contrasting data and comments which are not always consistent with the overall evaluation of the paper. The aim of providing a perception of headache as a social problem is obvious, but it is necessary to rewrite the discussion and evaluation of the information reported by the literature also in relation to the very different socio-economic and cultural situations that can influence headaches prevalence and their understanding . If this assessment is not made, it is not possible to evaluate the relevance of the data collected, and therefore the ability to stimulate health service decision-makers to take public health measures in relation to this disease will be reduced.

The English language is schematic, whereas when reasoning is developed, understanding is not immediate. A re-evaluation of the section "discussion" would be useful to make the article more readable and understandable.

Author Response

response attached in word document 

Reviewer 3 Report

This paper purports to measure the prevalence of headache but the methods as described would not allow such a claim.

The data were collected "using an anonymous survey disseminated through online platforms". The bias involved is obvious: people with headache are much more likely to have the interest to fill in a form about headache than those who do not suffer headache. This probably explains the fact that 75% of respondents were female. Migraine is known to be about 3 times more common in females than males and yet a higher proportion of males (48%) reported headache than females (43%). Presumably the males who did not suffer headaches were less likely to complete the questionnaire.

Headache was reported by 44% of those that completed the survey but this has little to do with the actual prevalence of headache in Makkah.

I have a number of specific quibbles:

Line 53: What does "Among those, 4.9 % are in the disability category" mean?

Line 54: "On the other hand, the prevalence of familial cluster headaches was found to be 6.2 % [16]." The paper in question points out that among those with cluster headache, it is familial in 6.2%. Quoting this paper out of context makes it sound as though 6.2% of the whole population has familial cluster headache, which is nonsense.

Lines 114-128 seem to have been copied from a set of instructions and should be removed.

Line 143: the following statement is incorrect. "Although there was no significant difference between sex and nationality and the prevalence of headaches, of the participants that reported headaches, 48.1% were male, 144 and 49.3% were non-Saudi (Table 1)." In fact Table 1 shows that 48.1% of males reported headaches and 49% of non-Saudis reported headaches. If you do the calculation on sex distribution, the correct figures are: of the 138+382 = 520 participants that reported headaches, 138/520 = 26.5% were male.

Line 164:  I think perhaps the authors are trying to say: "Despite the significant association observed between caffeine consumption and headache prevalence (p = .006), more than half of participants who drank caffeine (n = 479, 53.6%) did not report headaches; however, the amount of coffee drunk seems to be associated with headache (with 37.1% of non-caffeine drinkers, 43.8% of those who drank 1-2 cups per day, 50.0% of those drinking 3-4 cups and 57% of those drinking >4 cups per day reporting headaches) (Table 3)." If this is the intention, can they say whether the trend is statistically significant?

Line 177: in the discussion, the authors state "In contrast, migraine and TTH constituted 31.7% and 25.2% of the headaches reported in the current  study, respectively." This is the first time a distinction between headache types is mentioned? How were headache types diagnosed? Was this a self-reported diagnosis or was it verified from the responses in the questionnaire based on ICHD (or other) criteria? This is potentially useful data and should be presented under results (and the method of classification under methods).

Line 213: the statement "Chronic headaches can also weaken the immune system, leading to more frequent headache attacks and allergic reactions like asthma [39]." is not justified. Ref 39 makes it clear that there are associated co-morbidities but the causal relation is not clear.

Some of the errors quoted above may be due to inadequate English language skills rather than ignorance.

Author Response

replay attached in word document 

Round 2

Reviewer 1 Report

Dear Editor, the article is improved and can be publish in this form. 

Reviewer 2 Report

The authors have collected the suggestions and the work in its present form is acceptable for publication

English language is adeguate for publication in a scientific paper.

Reviewer 3 Report

Please remove the sentence:  "Additionally, results from previous literature showed that the prevalence of familial cluster headaches was 6.2% [16]." (Line 50). As previously explained it is wrong and misleading.

Line 154, "Although there was no significant difference between sex and nationality and the prevalence of headaches, of the participants that reported headaches, 48.1% were male,  and 49.3% were non-Saudi (Table 1)." is still wrong as previously pointed out.

Line 225 is still wrong.

Overall, my previous suggestions have been ignored.

The paper does not read badly at a superficial level but I suspect some of the errors referred to above may derive from poor understanding of English.